# Crystal Plasticity Parameter Optimization in Cyclically Deformed Electrodeposited Copper—A Machine Learning Approach

**DOI:** 10.3390/ma17143397

**Published:** 2024-07-09

**Authors:** Karol Frydrych, Maciej Tomczak, Stefanos Papanikolaou

**Affiliations:** 1NOMATEN Centre of Excellence, National Centre for Nuclear Research, Sołtana 7, 05-400 Otwock, Poland; karol.frydrych@ncbj.gov.pl (K.F.); maciej.tomczak@ncbj.gov.pl (M.T.); 2Institute of Fundamental Technological Research, Polish Academy of Sciences, Pawińskiego 5b, 02-106 Warsaw, Poland

**Keywords:** crystal plasticity, optimization, machine learning, long short-term memory networks, self-consistent modeling, Eshelby solution, cyclic deformation, low cycle fatigue

## Abstract

This paper describes an application of a machine learning approach for parameter optimization. The method is demonstrated for the elasto-viscoplastic model with both isotropic and kinematic hardening. It is shown that the proposed method based on long short-term memory networks allowed a reasonable agreement of stress–strain curves to be obtained for cyclic deformation in a low-cycle fatigue regime. The main advantage of the proposed approach over traditional optimization schemes lies in the possibility of obtaining parameters for a new material without the necessity of conducting any further optimizations. As the power and robustness of the developed method was demonstrated for very challenging problems (cyclic deformation, crystal plasticity, self-consistent model and isotropic and kinematic hardening), it is directly applicable to other experiments and models.

## 1. Introduction

Constitutive models enable the prediction of the behavior of materials subjected to external loadings from small [1] to large [2] length scales. In some straightforward problems, the constitutive models are simple, and their parameters are relatively easy to establish using standard experiments. For example, establishment of the Young’s modulus of an isotropic, linearly elastic material is unambiguous using just the result of a uniaxial tension test. However, in more complex situations, the constitutive models tend to use multiple non-linear equations and many parameters in order to capture the material’s behavior. Such a situation is present, e.g., in crystal plasticity, which is the topic of this paper.

The crystal plasticity (CP) theory in its most standard form accounts for the dislocation glide on strictly defined crystallographic planes. Therefore, the information about orientations of crystallites in a polycrystalline microstructure of a given material has to be provided. Besides taking into account these microstructural details, the hardening equations of crystal plasticity are typically similar to those of conventional plasticity and typically account for isotropic and kinematic hardening. Various optimization algorithms such as gradient optimization [3], the Newton–Raphson algorithm [4], the Levenberg–Marquardt method [5], Bayesian optimization [6], particle swarm optimization [7] and evolutionary algorithms [8,9,10,11,12,13,14,15,16,17,18,19] are used in order to establish the correct set of material parameters (see the introduction in [19] for a thorough discussion). However, all of them share a basic disadvantage: one has to repeat the optimization in the case of a new material or new test result.

In order to overcome this limitation, Wang et al. [20] developed a neural-network-based methodology for parameter optimization. The idea behind the approach is performing the training of a neural network (NN) once and then building the database of predicted outputs using the trained NN. The created database is used to predict the set of optimal parameters based on the provided input. Both the neural network and the database can be stored, and thus the main advantage of the proposed approach is that one does not need to perform a new optimization to obtain the parameters of a different material. A similar idea was presented in [21]. The main difference with regard to [20] is that machine learning (ML) was used only to establish reasonable parameter bounds, and then fine-tuning was performed using a genetic algorithm. Yet another approach to parameter optimization using ML was presented in [22], where the parameters of the model were found by training the NN to solve the inverse problem. This approach is clearly different from both [20,21] and it was demonstrated to work well for a model containing two parameters. Determining whether such an approach can work also for models with more parameters deserves further study, but it has to be stressed here that solving the inverse problem in the case of a *highly non-linear* problem such as the one presented here can be problematic.

In this article, we propose to perform the optimization for the case of the elasto-viscoplastic self-consistent (EVPSC) crystal plasticity model [14]. The investigated case is that of electrodeposited copper subjected to cyclic elasto-plastic deformation. We demonstrate that the proposed method is robust enough in order to provide correct predictions even in the highly complex problem as the one considered.

We build on the idea of using machine learning for the optimization of constitutive model parameters as described in [20]. However, there are important differences with regard to op. cit. First of all, we studied the case of a two-scale elasto-viscoplastic model, while the single-scale viscoelastic model was used in [20]. Second, we applied the LSTM approach directly to cyclic plasticity model data, while deep neural networks applied to data preprocessed using singular value decomposition technique were used in op. cit. Finally, single loading–unloading was considered in [20], while up to 50 cycles were simulated in the current study.

The article is structured as follows. After this introductory section, we describe the methodology (Section 2). This includes the description of the EVPSC model (Section 2.1) that was already thoroughly described in [14,23] and machine learning methodology (Section 2.2). These are followed by the optimization results using two approaches for feature extraction from stress–strain curves (Section 3). The discussion (Section 4) is related to the applicability of the proposed method to other non-obvious tasks. The paper ends with conclusions (Section 5).

## 2. Methodology

This section contains two subsections. The first one (Section 2.1) covers the description of the constitutive model together with the corresponding micromechanical scheme, which was used to provide input data for training the machine learning framework. The second one (Section 2.2) describes the machine learning approach applied for the purpose of parameter optimization in this paper.

### 2.1. The EVPSC Model

The sequential elasto-viscoplastic self-consistent (SEVPSC) code was recently developed and validated for the case of electrodeposited copper films [14] and is based on the sequential linearization scheme by [23]. (The code was written in FORTRAN and will be shared upon reasonable request. Researchers willing to use the code should contact Karol Frydrych.) The scheme coincides with the additive tangent approach [24] for the case of the Mori–Tanaka scheme. The results for the case of the self-consistent case were presented in [14,25]. Details of the self-consistent formulation can be found in the aforementioned papers. Nevertheless, some equations essential for the understanding of the present paper are provided in the following.

The strain rate, stress rate and stress have to fulfill the conditions stating that the average response over the polycrystal is consistent with the corresponding macroscopic values:(1)$$\dot{\bar{\varepsilon }}=\langle {\dot{\varepsilon }}_{g}\rangle \;,\;\dot{\bar{\sigma }}=\langle {\dot{\sigma }}_{g}\rangle \;,\;\bar{\sigma }=\langle \sigma_{g}\rangle \;.$$

In order to account for the interaction of the grain level and overall quantities, the additive tangent interaction law proposed by [26] or [27] is adopted:(2)$${\dot{\varepsilon }}_{g}-\dot{\bar{\varepsilon }}=-{\bar{M}}_{*}^{e}\cdot \left({\dot{\sigma }}_{g}-\dot{\bar{\sigma }}\right)-{\bar{M}}_{*}^{v}\cdot \left(\sigma_{g}-\bar{\sigma }\right)\;,$$
where $\dot{\bar{\varepsilon }}$, $\dot{\bar{\sigma }}$ and $\bar{\sigma }$ denote the overall strain rate, stress rate and stress. In Equation (Equation 2), ${\bar{M}}_{*}^{v}$ and ${\bar{M}}_{*}^{e}$ are the fourth-order inverse Hill tensors for a purely viscous problem and a purely elastic problem, respectively.

The parameters to be established with the proposed ML methodology concern the isotropic and kinematic hardening *at the level of individual grain.* In the following, the formulation of the single grain (subscript *g*) constitutive model is provided. The total strain rate is calculated by adding elastic ${\dot{\varepsilon }}_{g}^{e}$ and viscoplastic strain rates ${\dot{\varepsilon }}_{g}^{v}$ (the current version of the model is formulated in the small-strain theory):(3)$${\dot{\varepsilon }}_{g}={\dot{\varepsilon }}_{g}^{e}+{\dot{\varepsilon }}_{g}^{v}$$

In order to account for the elastic part, the anisotropic Hooke’s law is used:(4)$${\dot{\sigma }}_{g}=A_{g}^{e}\cdot {\dot{\varepsilon }}_{g}^{e}\;\mathrm{or}\;{\dot{\varepsilon }}_{g}^{e}=M_{g}^{e}\cdot {\dot{\sigma }}_{g}\;,$$
where $A_{g}^{e}$ is the fourth-order elastic stiffness tensor. The associated compliance tensor is $M_{g}^{e}={\left(A_{g}^{e}\right)}^{-1}$.

We use the rate-dependent CP formulation for a constitutive description at the single grain level. The plastic strain rate tensor at this level is calculated as a sum of shears on slip systems:(5)$${\dot{\varepsilon }}_{g}^{v}=\sum_{r=1}^{N}{\dot{\gamma }}^{r,g}P^{r,g}\;,\;\mathrm{where}\;P^{r,g}=\frac{1}{2}\left(m^{r,g}⊗n^{r,g}+n^{r,g}⊗m^{r,g}\right)$$
where ${\dot{\gamma }}^{r,g}$ is the slip rate, and $\left\{m^{r,g},n^{r,g}\right\}$ are the slip direction and the normal slip plane unit, respectively. Superscripts *r* and *g* denote a given slip systems and grain, respectively. $P^{r,g}$ is the symmetric Schmid tensor. The resolved shear stress (RSS) is a projection of the Cauchy stress tensor on the given slip system:(6)$$\tau^{r,g}=P^{r,g}:\sigma_{g}.$$

The slip rate depends on the RSS as well as on the slip system backstress $\chi^{r,g}$ and the critical resolved shear stress (CRSS) $\tau_{c}^{r,g}$ [28]:(7)$${\dot{\gamma }}^{r,g}={\dot{\gamma }}_{0}{\left|\frac{\tau^{r,g}-\chi^{r,g}}{\tau_{c}^{r,g}}\right|}^{n}\mathrm{sign}\left(\tau^{r,g}-\chi^{r,g}\right)\;.$$

Note that in Equation (Equation 7), both isotropic and kinematic hardening are included. The other quantities are the reference slip rate ${\dot{\gamma }}_{0}$ and the inverse of the strain rate sensitivity *n*. The tangent viscous compliance tensor is equal to a derivative of viscoplastic strain rate over the stress and thus equals:(8)$$M_{g}^{v}=n{\dot{\gamma }}_{0}\sum_{r=1}^{N}{\left|\frac{\tau^{r,g}-\chi^{r,g}}{\tau_{c}^{r,g}}\right|}^{n-1}\frac{\mathrm{sign}\left(\tau^{r,g}-\chi^{r,g}\right)}{\tau_{c}^{r,g}}P^{r,g}⊗P^{r,g}\;.$$

The evolution of the CRSS and backstress are governed by isotropic and kinematic hardening laws, respectively. The linear–exponential isotropic hardening evolution appears as follows:(9)$$\tau_{c}^{r,g}=\tau_{c}^{0}+h_{1}\gamma^{g}+\left(\tau_{\mathrm{sat}}-\tau_{c}^{0}\right)\left(1-e^{-h_{b}\gamma^{g}}\right),$$
where $\tau_{c}^{0},\tau_{\mathrm{sat}},h_{1},h_{b}$ are material parameters of the isotropic hardening and $\gamma^{g}=\int_{0}^{t}{\dot{\gamma }}^{g}dt$ is the accumulated shear in the grain. The Ohno–Wang kinematic hardening law [29] reads: (10)$${\dot{\chi }}^{r,g}=h_{k}{\dot{\gamma }}^{r,g}-h_{\mathrm{kb}}\left|{\dot{\gamma }}^{r,g}\right|{\left(\frac{\chi^{r,g}h_{\mathrm{kb}}}{h_{k}}\right)}^{h_{m}},$$
where $h_{k}$, $h_{\mathrm{kb}}$ and $h_{m}$ denote the kinematic hardening parameters. We have thus 4 isotropic hardening and 3 kinematic hardening parameters, which yields in total 7 hardening parameters subjected to optimization; cf. [14].

### 2.2. Machine Learning

Long short-term memory (LSTM) networks are a type of recurrent neural networks (RNNs) [30]. RNNs themselves are a type of artificial neural networks (ANNs) that are specifically designed for processing sequential data. The key difference of RNNs, as compared to traditional feedforward neural networks, is presence of connections that allow outputs to affect subsequent inputs of the same node in the network. The influence of history-related information on the current outputs can be thus considered. Therefore, RNNs were applied in areas where both order and context are of importance, e.g., language modeling and speech recognition [31,32]. Contrary to linear elasticity, models of plasticity are well known to be history- and path-dependent [33,34,35,36,37,38,39,40] and thus are naturally suitable for treatment with RNNs [41]. This is even more important in the case of cyclic loading, which is considered here [42]. Capturing time dependency is possible in RNNs through the hidden states that are updated at each time step when a new input is processed. The possible deficiencies of RNNs include vanishing or exploding gradients. Such problems were partially mitigated in more advanced variants such as LSTM (applied here) and Gated Recurrent Units (GRUs). These improvements brought the RNNs to a new level, enabling them to be a key tool in sequence modeling.

LSTM model architecture was selected because of its proven performance in other sequential prediction tasks, such as speech recognition and low parameter count when compared to fully connected networks, although as outlined in [43], there is no clear answer as to whether LSTM or GRUs are better at solving a given task. As compared to traditional RNNs, LSTM networks have a more complex architecture (cf. Figure 1). In particular, the LSTM cell contains several gates, such as the forget gate *f*, input gate *i*, cell gate *g* and output gate *o*. The presence of gates allows for the flow of information to be regulated; that is, the gates control which information is added to or removed from the cell state, enabling the network to maintain and access long-term dependencies more efficiently. Our implementation is a unidirectional LSTM as defined in PyTorch 2.0 [44] with a hidden state vector length of 2048. All three curves from the dataset are generated simultaneously by the network. The output is a tensor whose shape depends on the number of predicted values per cycle and number of cycles. Note that the term *tensor* in this subsection does not match the definition of tensor from the previous subsection and refers only to an understanding of this term in the machine learning jargon.

Two ways of evaluating the fitness of a given parameter set were tested. The first was exactly the same as in [14], i.e., the stress values at the points where loading changes its direction were compared; cf. Figure 2. This approach will be from now on termed App1. In this case, the shape of the output tensor is 2-by-100 (2 values in 2 points for each of the 50 cycles). In the second approach (App2), the stress–strain curves for each example were replaced with 3 simpler curves: stress amplitude *A*, mean stress $\bar{\sigma }$ and the slope of each cycle α in order to simplify the prediction task; cf. Figure 3. In this second case, the shape of the output tensor is 3-by-50. The input to the network is the same vector of seven parameters repeated 200 or 150 times (depending on the approach) to match the desired output sequence length.

Both input and output data were normalized to improve training stability by eliminating large differences in the scale of data. The model was trained for 300 epochs with a batch size of 16 and an initial learning rate of 1 × 10^−3^. The learning rate was decayed for each batch with a cosine annealing schedule. Warm restarts were applied after the first 50 epochs and after doubling the previous learning rate decay period after that until the end of training [46]. A weight decay of 1e-5 was used for regularization. The Adam optimization algorithm was applied to train the network with mean squared error as a loss function. The architecture of the network is shown in Figure 4.

## 3. Results

The method was applied as follows. First, the database of SEVPSC code results was created by generating 36370 loading–unloading curves. Each result was produced by running the SEVPSC code for the same loading scheme but with a different set of parameters. The parameters were generated by selecting every possible combination of parameters from a list of values. The list for each parameter was generated by selecting five equally spaced values from a set range (cf. Table 1). Then, part of the database was used for training the LSTM network, while the remaining part was moved to the test set (the ratio of training set size to the whole set size was equal to 60%). The MSE obtained in both approaches is shown in Figure 5.

The next step was thus using the trained LSTM network for parameter optimization. The approach consists of using the trained LSTM to generate predictions in points lying in the parameter space that are more densely distributed than the set used for training. For this, 35.8 million results were generated. Then, for the reference case (obtained by running the SEVPSC code with random parameters), the closest result in the LSTM-generated database is searched by calculating the mean squared error (MSE) between reference case curves and generated examples and selecting the generated example with the lowest MSE. We have tested the method for 30 different parameter sets. The six representative parameter sets obtained using this technique are shown in Table 2. The corresponding stress–strain curves obtained using the optimized parameters compared against the reference ones are shown in Figure 6. The entire group of parameter sets is presented in the Appendix A. We divided the results into six categories:1.Very good or reasonable agreement of SS curves obtained using parameters optimized in both approaches—Figure 6a and Appendix A;2.Disagreement in the first cycle and reasonable agreement of SS curves obtained using parameters optimized in both approaches—Figure 6b and Appendix A;3.Reasonable agreement of SS curves obtained using parameters optimized in App 1 (lack of convergence for App 2 parameters)—Figure 6c and Appendix A;4.Reasonable agreement of SS curves obtained using parameters optimized in App 2 (lack of convergence for App 1 parameters)—Figure 6d and Appendix A;5.Striking disagreement or lack of convergence—Figure 6e and Appendix A;6.Lack of convergence for the optimized parameters in both approaches—Figure 6f and Appendix A.

Out of 30 cases examined, 13 cases show very good or reasonable agreement in both approaches (category 1). Three cases show reasonable agreement in late cycles but disagreement in the first cycle (category 2). It can be thus stated that more than half of the cases the method worked correctly with both approaches. In four cases, App 1 provided the correct prediction, while App 2 did not converge (category 3). The reverse situation (category 4) was present in another four cases. Therefore, in 24 cases (80%), at least one approach provided a reasonably correct set of parameters. Concerning category 5, in one case, both approaches provided parameter sets, leading to striking disagreement in SS curves (Figure 6e and Appendix A). In another case (cf. Appendix A), there was striking disagreement for App 2 and a lack of convergence for App 1. Finally, in four cases, neither approach provided a parameter set that can converge in actual SEVPSC simulation (category 6).

Let us now take a closer look on the obtained results. In terms of the approaches, it seems that both of them are balanced, and we cannot argue that either App1 or App2 is better. Cases where the predicted SS curve agrees with the actual one (Category 1, cf. Figure 6a and S1) prove the potential of the method. The fact that almost half of cases show only slight deviations in the shape of the curve in the first cycle is actually slightly surprising as the information about the shape of the first cycle was absent in both methods. It is thus easy to understand why three other cases (Category 2, cf. Figure 6b and Appendix A) agree with the reference curve in the cyclic deformation regime while having some deviation in the first cycle. Moreover, these three cases can be treated as correct predictions since the LSTM had no information about the shape of the initial loading curve. In the rest of the cases (Categories 3–6), there was either agreement or a lack of convergence, except for the curve shown in Figure 6e. Cases where the simulations with optimized parameters fail to converge are not useful but also harmless. Thus, only 1 case out of 30 led to a misleading prediction.

## 4. Discussion

The trained network was used for parameter identification. But it could be also used as a surrogate model to obtain the same overall information in a far shorter time. Such an approach was already presented in a number of papers; e.g., in [47], the surrogate ML model was trained using data from discrete dislocation dynamics. The network was able to predict the stress–strain curve based on the initial dislocation configuration. On the other hand, Deshpande, Lengiewicz and Bordas [48] trained their deep learning (DL) network based on large deformation finite element method (FEM) simulations. Such an approach made it possible to predict the deformed shape based on the initial specimen geometry and applied boundary conditions.

Many surrogate models were also constructed in the framework of crystal plasticity (CP). A surrogate model for the viscoplastic self-consistent (VPSC) model was built in [49]. The researchers developed both forward and backward ML configurations. The first made it possible to predict the final crystallographic texture and stress–strain curve based on the initial texture and hardening parameters. The second enabled the prediction of the initial texture and hardening parameters based on the final texture and stress–strain curve. A neural network allowing the prediction of the stress–strain curve and texture but this time trained using the crystal plasticity finite element method (CPFEM) was presented in [50]. CPFEM-trained ML architecture was used to predict cyclic stress values in low-cycle-fatigue experiments in [51]. An interesting contribution was presented in [52], where the ML was trained on an elastic–viscoplastic fast Fourier transform (EVPFFT) model, and the prediction concerned spatially resolved crystallographic orientations. A novelty in [53] was the application of CP-trained ML to non-monotonic strain paths. In [54], it was demonstrated that using an encoder–decoder deep learning framework, it is possible to regenerate stress–strain curves for a material subjected to several loading–unloading cycles.

We hereby stress that even though our approach of using the ML architecture for parameter optimization could be used as a surrogate model, it is currently clearly different from the aforementioned approaches. In our case, the main purpose of training the ML network was to find parameters for the physical model (EVPSC), which still serves as a base of all the simulations provided. Moreover, constructing a surrogate model that estimates only the stress–strain points at loading direction changes (App1) or features of each cycle (App2) would be pointless. Even if the surrogate model is able to predict the full stress–strain curve (and our trained ML framework *is not*), it would still reduce the wealth of information that can be attained from the EVPSC model. As pointed out in [14], the main advantages of the EVPSC model are its capabilities to account not only for the global response, but also to give insight into details of the plastic deformation occurring at every grain. Such information would be lost in the case of constructing the surrogate model that returns only the average quantities.

## 5. Conclusions

This paper demonstrates the possibility of using a machine learning framework based on LSTM to efficiently optimize material parameters in a highly complex situation of modeling the cyclic deformation of copper films using the elasto-viscoplastic self-consistent model with isotropic and kinematic hardening. One can conclude that the method is able to provide a reasonably accurate estimate of the parameter set that in most cases provides the correct cyclic stress–strain.

A significant advantage of the proposed machine-learning-based method over optimization algorithms is that in the case of a different material subjected to the same loading conditions, one does not need to perform any other simulations to obtain the optimal parameters. One could argue that a similar result could be obtained using only a sufficient database. Using such a database instead of the machine learning approach, however, has two disadvantages. First, in order to achieve an accuracy comparable to the one attained here, one should perform about *10 thousand* times more simulations (because the number of simulations used for training was 3670 and the number of data points generated by trained ML framework was 35.8 million). Second, in order to use the database for future purposes, one would have to store the whole database of results (which would be 35.8 million times 977 kB = 35 TB), while in our case it is enough to store the trained model (which would be just 50.8 MB) and when necessary regenerate the database with an arbitrarily large or small number of example inputs to achieve a desired database size.

In this paper, the LSTM-based architecture was applied, as some advantages of LSTM over more traditional recurrent units were demonstrated [43]. However, op. cit. also stated that the results of comparison between LSTM and GRUs are not conclusive and shall depend on the application. Moreover, new architectures specifically designed for mechanics problems were proposed; cf., e.g., the minimal state cells (MSCs) [55]. In the future, it will be very interesting to perform detailed studies where the efficiency of various architectures, like LSTM, GRUs, and MSCs (and possibly others) is compared. This task is, however, surely outside the scope of the present contribution.

Of course, the developed approach is not able to provide the material parameters based on different loading conditions than were used as an input. Therefore, in such a case, one would have to repeat the optimization. The method was demonstrated for a highly challenging problem of cyclic deformation simulated with the micromechanical model accounting for both isotropic and kinematic hardening. It is expected that the approach will be directly applicable to other cases. In particular, one can naturally apply the developed methodology to a case of studying low and high cycle fatigue with plasticity and damage models; cf., e.g., [56]. 

## Figures and Tables

**Figure 1 materials-17-03397-f001:**
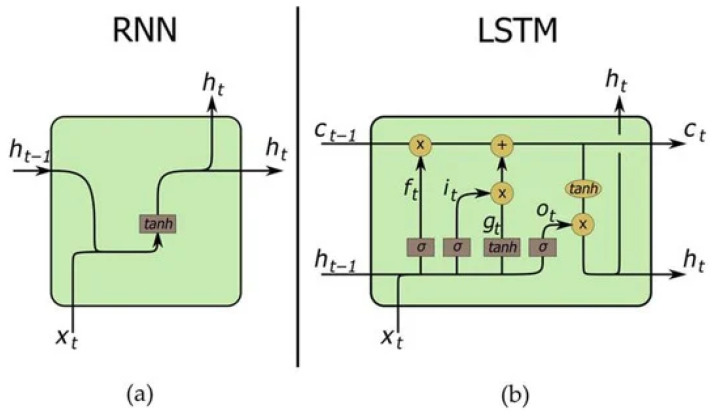
Scheme showing difference between the architecture of the typical (**a**) recurrent neural network (RNN) and (**b**) long short-term memory (LSTM) network. The figure was reprinted from [45] based on the Creative Commons Attribution (CC BY) license.

**Figure 2 materials-17-03397-f002:**
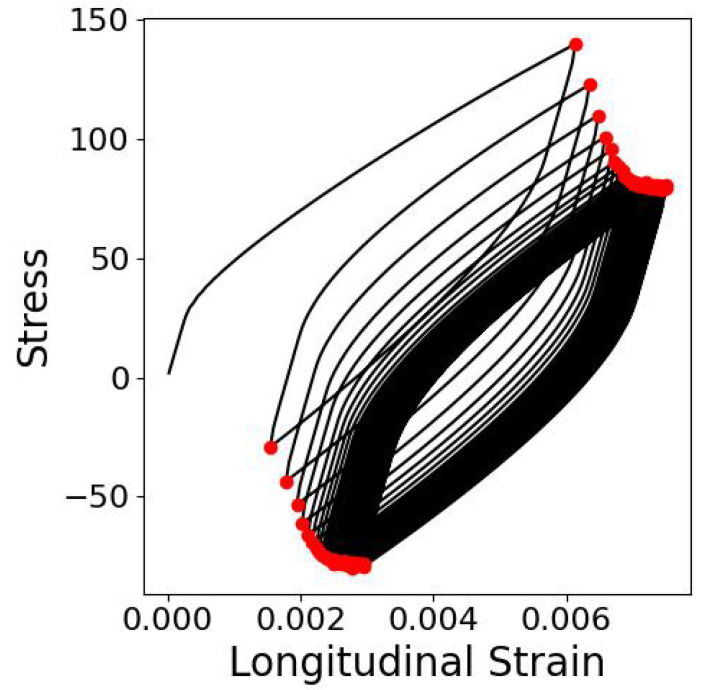
Scheme showing locations of difference evaluations according to Approach 1.

**Figure 3 materials-17-03397-f003:**
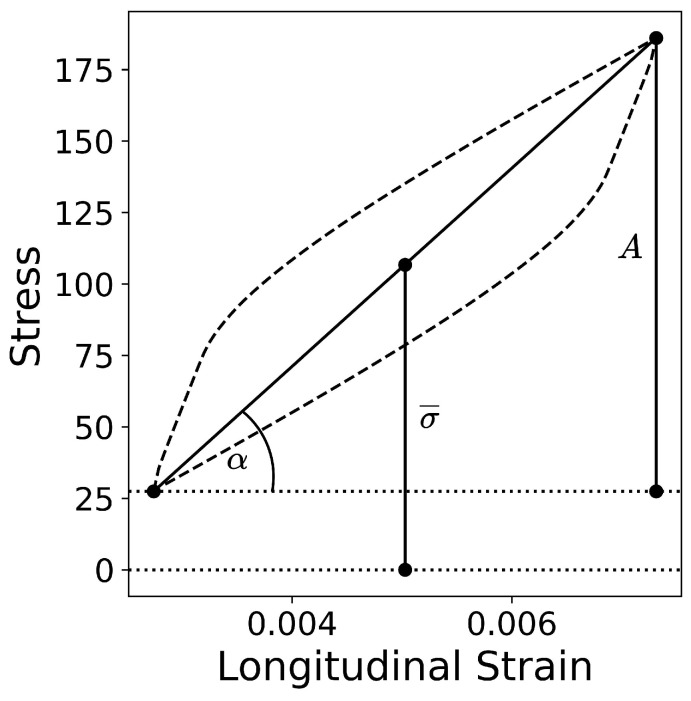
Scheme showing difference evaluations according to Approach 2: stress amplitude *A*, mean stress $\bar{\sigma }$, and slope of each cycle tan(α) were used.

**Figure 4 materials-17-03397-f004:**
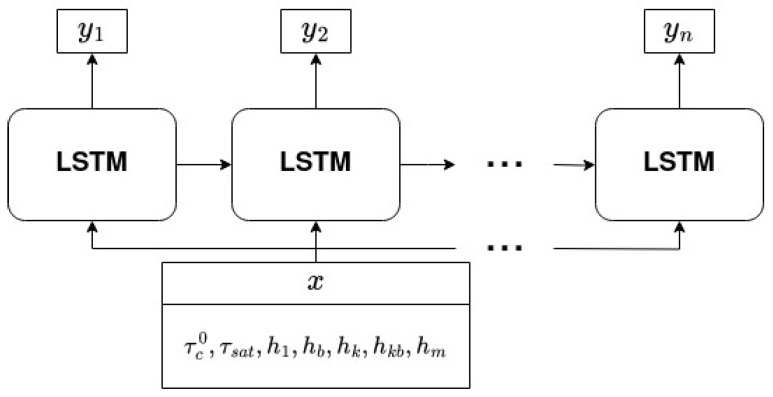
Scheme showing the architecture of the network used in the present contribution. The input *x* is a set of 7 hardening parameters; outputs $y_{i}$ are data describing the shape of stress–strain curves.

**Figure 5 materials-17-03397-f005:**
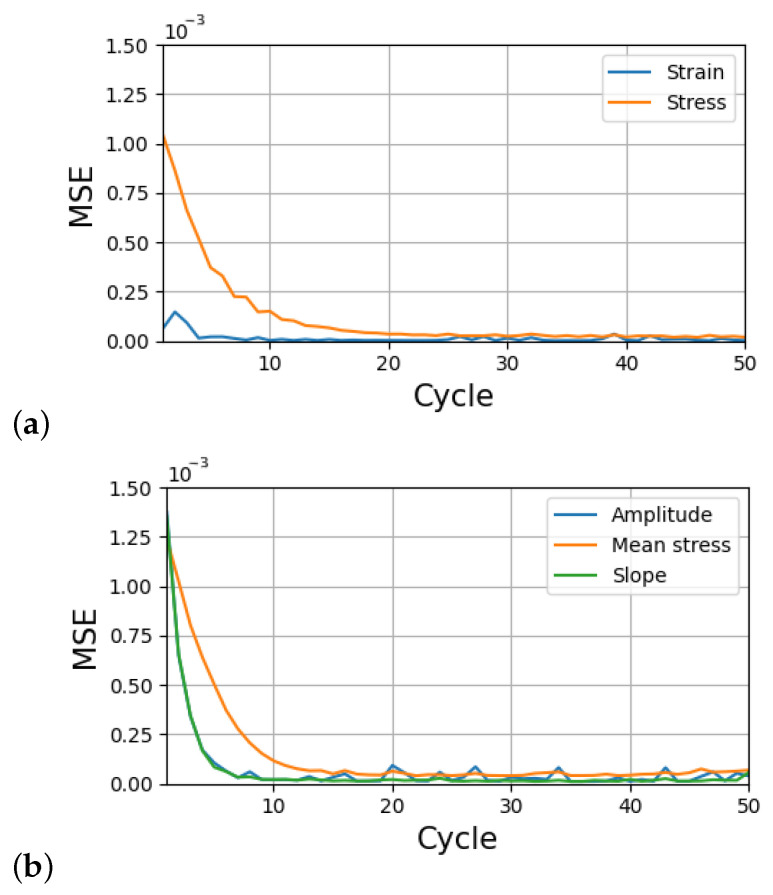
Mean squared error as a function of cycles in (**a**) approach 1 and (**b**) approach 2.

**Figure 6 materials-17-03397-f006:**
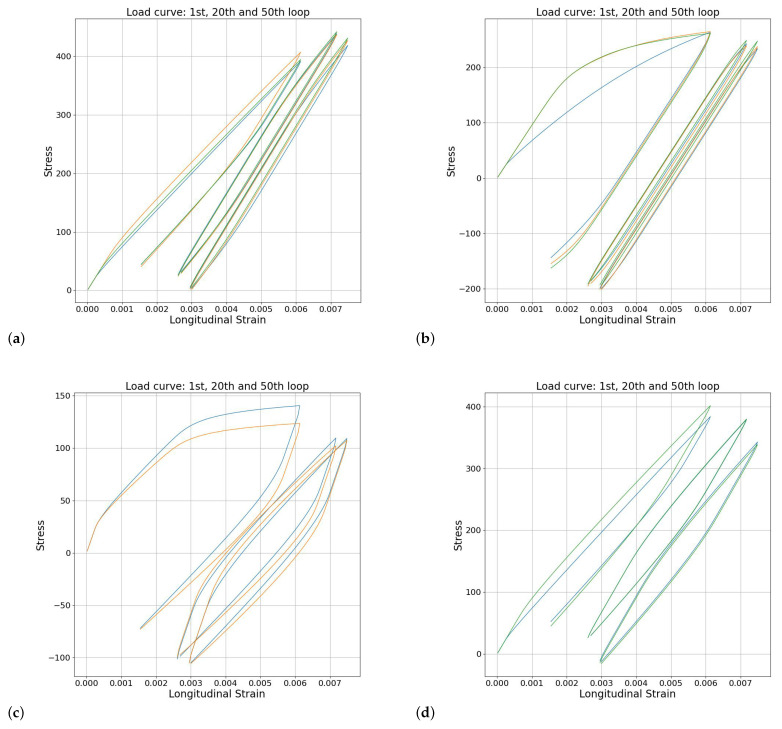
The results of using neural networks for optimization: stress–strain curves obtained using the SEVPSC code for the arbitrary parameter set and its closest neighbors. Only the first and last cycle are shown for the sake of clarity. The selected examples refer to categories: (**a**) 1: very good or reasonable agreement of SS curves obtained using parameters optimized in both approaches, (**b**) 2: disagreement in the first cycle and reasonable agreement of SS curves obtained using parameters optimized in both approaches, (**c**) 3: reasonable agreement of SS curves obtained using parameters optimized in App 1 (lack of convergence for App 2 parameters), (**d**) 4: reasonable agreement of SS curves obtained using parameters optimized in App 2 (lack of convergence for App 1 parameters), (**e**) 5: striking disagreement or lack of convergence, (**f**) 6: lack of convergence for the optimized parameters in both approaches.

**Table 1 materials-17-03397-t001:** Ranges of the hardening parameters serving to generate the database.

	$\tau_{c}^{0}$	$\tau_{\mathrm{sat}}$	$h_{1}$	$h_{b}$	$h_{k}$	$h_{\mathrm{kb}}$	$h_{m}$
Min	10.0	10.0	0.0	0.0	0.0	0.0	0.0
Max	80.0	120.0	5.0	120.0	1.0$\cdot 10^{5}$	1000.0	10.0

**Table 2 materials-17-03397-t002:** Reference parameters vs. parameters that made LSTM generate closest matching curves.

	$\tau_{c}^{0}$	$\tau_{\mathrm{sat}}$	$h_{1}$	$h_{b}$	$h_{k}$	$h_{\mathrm{kb}}$	$h_{m}$
Category 1							
Reference	1.00$\;\times \;10^{1}$	6.50$\;\times \;10^{1}$	4.17	4.00$\;\times \;10^{1}$	1.00$\;\times \;10^{5}$	5.00$\;\times \;10^{2}$	3.33
App 1	2.91$\;\times \;10^{1}$	9.00$\;\times \;10^{1}$	4.09	1.09$\;\times \;10^{1}$	1.00$\;\times \;10^{5}$	4.55$\;\times \;10^{2}$	2.73
App 2	1.64$\;\times \;10^{1}$	8.00$\;\times \;10^{1}$	1.36	2.18$\;\times \;10^{1}$	1.00$\;\times \;10^{5}$	4.55$\;\times \;10^{2}$	2.73
Category 2							
Reference	1.00$\;\times \;10^{1}$	8.33$\;\times \;10^{1}$	8.33$\;\times \;10^{-1}$	1.20$\;\times \;10^{2}$	3.33$\;\times \;10^{4}$	1.00$\;\times \;10^{3}$	0.00
App 1	6.73$\;\times \;10^{1}$	1.00$\;\times \;10^{2}$	5.00	1.09$\;\times \;10^{1}$	9.09$\;\times \;10^{3}$	5.45$\;\times \;10^{2}$	9.09$\;\times \;10^{-1}$
App 2	6.73$\;\times \;10^{1}$	1.20$\;\times \;10^{2}$	4.55$\;\times \;10^{-1}$	1.09$\;\times \;10^{1}$	9.09$\;\times \;10^{3}$	8.18$\;\times \;10^{2}$	3.64
Category 3							
Reference	1.00$\;\times \;10^{1}$	1.00$\;\times \;10^{1}$	1.67	1.20$\;\times \;10^{2}$	3.33$\;\times \;10^{4}$	8.33$\;\times \;10^{2}$	8.33
App 1	1.00$\;\times \;10^{1}$	1.00$\;\times \;10^{1}$	5.00	2.18$\;\times \;10^{1}$	2.73$\;\times \;10^{4}$	8.18$\;\times \;10^{2}$	7.27
App 2	1.64$\;\times \;10^{1}$	1.00$\;\times \;10^{1}$	5.00	2.18$\;\times \;10^{1}$	2.73$\;\times \;10^{4}$	7.27$\;\times \;10^{2}$	3.64
Category 4							
Reference	1.00$\;\times \;10^{1}$	2.83$\;\times \;10^{1}$	5.00	8.00$\;\times \;10^{1}$	1.00$\;\times \;10^{5}$	5.00$\;\times \;10^{2}$	3.33
App 1	4.18$\;\times \;10^{1}$	2.00$\;\times \;10^{1}$	1.82	2.18$\;\times \;10^{1}$	1.00$\;\times \;10^{5}$	4.55$\;\times \;10^{2}$	2.73
App 2	2.91$\;\times \;10^{1}$	1.20$\;\times \;10^{2}$	1.36	0.00	1.00$\;\times \;10^{5}$	4.55$\;\times \;10^{2}$	2.73
Category 5							
Reference	1.00$\;\times \;10^{1}$	1.00$\;\times \;10^{1}$	5.00	1.20$\;\times \;10^{2}$	3.33$\;\times \;10^{4}$	6.67$\;\times \;10^{2}$	0.00
App 1	1.00$\;\times \;10^{1}$	1.00$\;\times \;10^{1}$	5.00	2.18$\;\times \;10^{1}$	2.73$\;\times \;10^{4}$	9.09$\;\times \;10^{2}$	8.18
App 2	22.73	30.00	5.00	32.73	0.00	0.00	0.00
Category 6							
Reference	10.00	83.33	1.67	60.00	0.00	1000.00	0.00
App 1	35.45	90.00	2.73	43.64	0.00	1000.00	4.55
App 2	22.73	90.00	0.91	54.55	0.00	727.27	0.00

## Data Availability

The original contributions presented in the study are included in the article/Appendix A, further inquiries can be directed to the corresponding author.

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
