# Peer review of "Crystal Plasticity Parameter Optimization in Cyclically Deformed Electrodeposited Copper—A Machine Learning Approach"

_materials, 2024, doi:10.3390/ma17143397_

Round 1

Reviewer 1 Report

Comments and Suggestions for Authors

In this manuscript, the authors evaluated the effectiveness of machine learning in the prediction of mechanical parameters of new materials. The machine learning was applied at complex models and problems, including cyclic deformation, crystal plasticity, self-consistent model, isotropic and kinematic hardening.

The results of this work could potentially enhance and spread the application of machine learning in other, similar problems and models. This work has good novelty, and the results are well organized. I would like to recommend publication without any improvements.

Reviewer 2 Report

Comments and Suggestions for Authors

In this manuscript, the author used ML to optimize the constitutive model parameters for the existing elasto viscoplastic self-consistent (EVPSC) crystal plasticity model. The proposed method has been proven to be very stable through the example of copper electrodeposition under cyclic elastic-plastic deformation. The algorithm considers a two scale elasto viscoplastic model and directly applies the LSTM method to the cyclic plastic model, with up to 50 cycles of data. This optimization method can also obtain accurate predictions in dealing with complex problems such as cyclic deformation, crystal plasticity, self-consistent models, isotropy, and motion hardening. This method can predict the properties of different materials under the same loading conditions without the need for any other simulations to obtain the optimal parameters. The research results show that this is clearly due to the results of existing algorithms. The research has strong innovation and provides a new and effective method for studying the plasticity of material crystals. The theoretical analysis, numerical implementation, and analytical discussion of the manuscript are all very detailed and reasonable. The data and charts in the obtained paper and supporting materials are abundant, and the analysis is appropriate, which can effectively support the conclusions drawn.So I recommand to accept the manuscript for publication as it is.

Reviewer 3 Report

Comments and Suggestions for Authors

In this paper, the problem of cyclic deormatization of electrodeposited copper is investigated and parameter optimization is performed using machine learning methods, namely recurrent neutron networks. The topic of the paper is interesting, as parameter optimization even for simple plasticity models can be a challenge, but the results obtained raise some questions.

1. Insufficient description of the recurrent neural network architecture in 2.2, given that the physical model in 2.1 is described in detail, and most of the paper is devoted specifically to the use of machine learning.

2. There is no justification for choosing an LSTM architecture that contains more gate functions and additional feedback for internal memory state [Gang Chen. A Gentle Tutorial of Recurrent Neural Network with Error Backpropagation]. If the authors say they chose the recurrent network because of fewer parameters in these type of neural networks, why not choose a gated recurrent unit (GRU) architecture with fewer gate functions and one feedback? Or take the Minimal State Cells architecture proposed by Prof. Mohr specifically designed to describe path-dependent plasticity in metals with even fewer parameters [C. Bonatti, D. Mohr One for all: Universal material model based on minimal state-space neural networks] ?

3. How relevant is it to predict only the shape of stress-strain curves based on model parameters? Since in reality one is often interested of values of physical model parameters based on experimental data. For example, this paper [V.V. PogorelkoA.E. Mayer, E.V. FominE.V. Fedorov  Examination of machine learning method for identification of material model parameters] shows the solution of the direct (prediction of the shock wave velocity profile based on physical model parameters) and inverse problem (prediction of the physical model parameters based on the  shock wave profile) using a feed-forward propagation neural network for shock-wave loading in metals. In my opinion, this paper lacks this part of the work.

Round 2

Reviewer 3 Report

Comments and Suggestions for Authors

The authors have corrected all the shortcomings I expressed in the article, I recommend this version of the article for publication